# Road Pavement Structural Health Monitoring by Embedded Fiber-Bragg-Grating-Based Optical Sensors

**DOI:** 10.3390/s22124581

**Published:** 2022-06-17

**Authors:** Janis Braunfelds, Ugis Senkans, Peteris Skels, Rims Janeliukstis, Jurgis Porins, Sandis Spolitis, Vjaceslavs Bobrovs

**Affiliations:** 1Communication Technologies Research Center, Riga Technical University, LV-1048 Riga, Latvia; ugis.senkans@rtu.lv (U.S.); sandis.spolitis@rtu.lv (S.S.); 2Institute of Telecommunications, Riga Technical University, LV-1048 Riga, Latvia; jurgis.porins@rtu.lv (J.P.); vjaceslavs.bobrovs@rtu.lv (V.B.); 3Department of Roads and Bridges, Riga Technical University, LV-1048 Riga, Latvia; peteris.skels_1@rtu.lv; 4Institute of Materials and Structures, Riga Technical University, LV-1048 Riga, Latvia; rims.janeliukstis_1@rtu.lv

**Keywords:** fiber Bragg grating (FBG), fiber optical sensors (FOS), structural health monitoring (SHM), strain measurements

## Abstract

Fiber Bragg grating (FBG) optical sensors are state-of-the-art technology that can be integrated into the road structure, providing real-time traffic-induced strain readings and ensuring the monitoring of the road’s structural health. By implementing specific FBG sensors, it is possible to detect each vehicle’s axle count and the induced strain changes in the road structure. In this study, FBG sensors are embedded at the top of the 240-mm-thick cement-treated reclaimed asphalt pavement mixture layer of the road (specifically, 25 mm deep within the road). Optical sensors’ signal interrogation units are used to measure the strain and temperature and collect data of the road’s passing vehicles, starting from passenger cars that have two axles and up to heavy trucks that have six axles. Passenger cars with 2 axles generate a typical (90% events) strain of 0.8–4.1 μm/m, the 2-axle minibus 5.5–8.5 μm/m, 2–3-axle trucks 11–26 μm/m, but 4–6-axle trucks 14–36 μm/m per each axle. A large number of influencing parameters determine the pavement design leading to the great uncertainty in the prediction of the strain at the boundary between the asphalt surface and cement-treated base layers. Real-time strain and temperature measurements help to understand the actual behavior of the pavement structure under an applied load, thus assisting in validating the proposed pavement design.

## 1. Introduction

Traffic volume on the Latvian road network had been continually increasing until 2019, when the average annual daily traffic (AADT) reached 6525 vehicles in 24 h. Even though the AADT (based on traffic counting data) decreased by 7% in 2020 [1], the general trend shows an increase in overall traffic volumes and road loading by equivalent 10-ton axle loads.

With the rapid growth of worldwide fiber optical networks, all the related technology has also advanced and been adopted for industry purposes. One of such perspective technologies is fiber optical sensors (FOS). FOS have multiple advantages such as an immunity to corrosion [2,3,4] and electromagnetic interference [3,4,5,6,7], the capability for distributed and long-distance measurements [8,9], harsh environment and high-temperature durability [10], high precision [8,11], easy integration [8,11] and high sensitivity [3,9,12], which can be efficiently utilized for necessary applications. 

FBG technology allows for the relatively easy and convenient implementation of FOS into a necessary architecture. FBG sensors are versatile in the matter of the parameters to be measured; for instance, strain [13,14], temperature [9,15], pressure [16], humidity [9,17], and many more can be measured and monitored. With regard to industry, FBG FOS realization is topical in structural health monitoring (SHM) applications for dams [10,18], pipelines [10,16], slopes [10], tunnels [10,19], bridges [10,20], railroads [21], roads [14,22], and so on. 

The changes in traffic volumes and road loading have a direct impact on the rate of deterioration of the road pavement. SHM applications, by incorporating FOS inside the pavement, are a valuable tool for infrastructure managers. SHM tools assist in the planning of reconstruction and maintenance smartly and justifiably, based on real-time data. Such tools help in making appropriate decisions for planning the maintenance and reconstruction works and validate the design assumptions by measuring actual pavement structural layers’ responses under the load conditions. Since new materials and structural layers are introduced, such as modified new types of asphalts, stabilized base layers, cement-treated reclaimed asphalt pavement (RAP) mixture layers, etc., the increased importance for such solutions needs to be researched. 

Our research process is structured into three main parts—preparation, the calibration of the implemented FOS with a falling weight deflectometer (FWD), and the experimental study of real transport traffic and strain measurements. Further paper sections are organized as follows. Construction and analysis of the experimental setup are described in Section 2. Section 3 discusses the FWD application method for calibration purposes with regard to the operation of FBG optical sensors, while Section 4 focuses on real-time transport traffic strain monitoring for SHM by using FBG optical sensors that are integrated within road infrastructure. In the end, the results are concluded, and the effectiveness of the proposed technique is evaluated.

## 2. The Working Principle of Our Optical Sensor Signal Interrogation Unit and FBG Optical Sensors

During the whole research process, it is necessary to use the authors’ previously individually developed [14] mobile optical sensors signal interrogation unit (OSSIU), which is suitable for field use to provide different sorts of strain measurements in real-time (including induced strain by real-time transport traffic). The main parameters of this device are described in Table 1. 

The OSSIU unit consists of a superluminescent emitting diode (SLED) (parameters mentioned in further paragraphs), an optical circulator (OC) necessary for the separation of the directions of the optical signals, as well as a spectrometer and digital signal processor (DSP) to investigate the received optical spectrum and data from the FBG optical sensors.

As for the optical sensors, a pair of commercial FBG strain and temperature sensors were chosen. A temperature sensor (Xi’an Aurora Technology Co, Xi’an, China) is necessary to ensure the FBG optical strain sensor’s calibration during the whole research period, thus excluding the ambient temperature’s effect on the received strain values. The main parameters of the used optical FBG sensors are summarized in Table 2.

The experimental working principle of our FBG strain sensor is depicted in Figure 1. Here, for an optical input signal, a high-power SLED broadband light source with a central wavelength of 1550 nm and a 3-dB bandwidth of 55 nm is chosen and has already been integrated inside our OSSIU unit.

The optical spectrum of the input signal is visible in Figure 1a. The optical signal is transmitted through the optical fiber as shown in the middle-upper part of Figure 1, where the FBGs sensors are inscribed. The reflected optical signal of the FBG sensor at the reference wavelength can be seen in Figure 1b. This reflected optical spectrum of the strain sensor is processed and observed in our mobile OSSIU unit. Figure 1c, on the other hand, shows the transmitted optical signal after passing the FBG strain optical sensor.

Integration of optical FBG sensors within the road architecture during the construction process allows for monitoring of the reflected optical sensors’ wavelength or frequency changes induced by the strain in the long term. In this research, the strain shift is generated with either a falling weight deflectometer or real-time transport traffic (vehicles passing over the FBG sensors). The induced strain and its shift can provide information about the vehicles or objects on top of the road structure and, no less important—the integrity or structural health of the road construction at that specific time. The strain shift is calculated (see Table 3 for the received strain data with the FWD experiment and Table 4 for real-time received transport traffic strain data) in this research using an equation [23]:(1)$$\Delta \varepsilon =\frac{\Delta \lambda -B\times \Delta T}{A\times \Delta l}+\left(\Delta T\times CTE\;\times \Delta l_{2}\right)$$
where:$\Delta \lambda =\frac{\lambda_{act}-\lambda_{0}}{\lambda_{0}}$$\Delta T=\left(T_{act}-T_{0}\right)$$\Delta l=\frac{l{\;}_{FAL\;}}{l_{FFL}}$$\Delta l_{2}=\frac{\left(l_{FAL}-l_{FFL}\;\right)}{l_{FAL}}$Δε—Strain shift (µm/m);$\lambda_{0}—$Initial strain wavelength (nm);$\lambda_{act}—$Actual strain wavelength (nm);$T_{0}—$Initial temperature (°C);$T_{act}—$Actual temperature (°C);$l_{FAL}—$Anchoring length (m);$l_{FFL}—$Free fiber length (m);CTE—Coefficient of thermal expansion [μ·ε·°C];*A*—Coefficient [$\mu \cdot \varepsilon^{-1}$];*B*—Coefficient [°$C^{-1}$];

Similar to the strain shift calculation, it is possible to calculate wavelength versus temperature shifts by using an equation:(2)$$\Delta T=\frac{\lambda_{act}-\lambda_{0}}{k}$$
where:ΔT= Temperature shift (°C);$\lambda_{act}=$ Actual wavelength (nm);$\lambda_{0}=$ Initial wavelength at time “0” (nm);*k* = Monomial coefficient $\left(\frac{pm}{{}^{\circ}C}\right)$.

## 3. Integration of the FBG Sensors into Asphalt Pavement

Once the setup and all the preparation work have been completed, the location for the experiment is chosen and carried out in a construction site, where FBG optical sensors are embedded. FBG optical sensors are integrated into the road with geographical locations, Latvia. Available online: https://google/maps/hkxm4er4slcqcaff7 (accessed on 24 May 2022). Here a contracting company, “Binders” Ltd., (Riga, Latvia) and a joint stock company, “ACB”, (Riga, Latvia), was performing reconstruction on the road of national importance (A8 Riga-Jelgava-Meitene border km 19.20–29.95 km). This particular road connects Riga city (capital of Latvia) with the border of Lithuania.

Firstly, an experimental setup (please see Figure 2) for real-time transport traffic strain monitoring is developed to display the overall method for the investigation of SHM applications. 

As can be observed in Figure 2, the FBG strain and temperature sensors are embedded in the upper part of the 240-mm-thick cement-treated RAP mixture layer, specifically, 25 mm deep inside the cement-treated RAP mixture layer, that is, under 35-mm-thick SMA11 (stone mastic asphalt), 60-mm-thick AC22 and 100-mm-thick AC32 (asphalt concrete) layers and monitored remotely with OSSIU. It detects all the induced strain (and thus reflected wavelength/frequency shifts of the FBG optical sensor) by the vehicles passing the upper layer of the road. Unlike our previous research [14], where FBG optical sensors were embedded within the temporary road—between 30-mm-thick SMA11 and 60-mm-thick AC11 layers—in this research, we have chosen a different approach by studying the embedment of deeper FBG optical sensors in the cement-treated RAP mixture layer (~220 mm from the surface, see the left side of Figure 3) and strain measurement detection, and hence also test such technologies’ performance in permanent road infrastructure. In this research, new types of optical sensors are also chosen, providing a more reliable embedment realization in road infrastructure.

FBG strain and temperature sensors are integrated into a position so that the majority of passing transportation would almost certainly drive over that specific position. Optical sensors and all of their connecting armored optical patches are inserted into sawn grooves (visible on the left side of Figure 3). The width of the groove where FBGs optical sensors are inserted is 60 mm, and its depth is 50 mm, while the part of the groove where only optical patches are inserted is reduced to 35 mm, as it is not necessary for a wider groove in that position. Connecting optical fibers of the FBG sensors are inserted within the heat-resistant and mechanically durable pipe with a diameter of 30 mm. This is done to provide for the longevity of the FBG optical sensors and their fibers.

Some of the main difficulties that can be encountered and need to be dealt with during the embedment process are firstly choosing a specifically suitable road embedment type of strain FBG sensor, as some of the FBGs that are attached to the material’s surface might not position well. Secondly, additional optical fibers’ protection from the mechanical influence (as the asphalt laying machine was driving over the embedment position during the construction process) and temperature influence (which was increased as the hot asphalt layer was poured on) has to be completed as discussed in the previous paragraph. Last but not least, it was also important to ensure that the FBG sensors were embedded so that there would not be any air gaps between the surface and the layers on the road under the position where the FBGs were placed.

Once the sensors are installed and the construction process of the road infrastructure is finished, it is possible to use the FWD device, calibrate the measurements of the optical FBG sensors, measure the data acquired, and start the real-time transport traffic monitoring for comprehensive data acquisition. The performed experiments are discussed in the next sections of this article.

## 4. The Initial Experiments and Calibration Process of the Embedded FBG Sensors 

FWD applications (see Figure 4) for evaluation of the collected strain data are necessary to ensure proper installment of the realized FBG optical sensors. To make certain that the sensors’ response data are correct, it is necessary to calibrate the acquired strain data concerning the related timeline. FWD is widely used [24,25,26] for non-destructive road pavement testing and as a research test tool. Mostly, FWD gathered data are used to estimate the stiffness parameters of the underlying pavement structure. A single pulse load of 50 kN generates about 707 kPa of pressure (which is relatively equivalent to the amount of pressure generated by the heavy type of passing trucks—equivalent to a 10-ton axle load) under the load plate and is depicted in Figure 5, where load versus time data can be observed. Such parameters are typical to the European practice for road infrastructure testing, thus allowing us to ensure that FOS applications in road SHM complies with the road-building industry’s needs and standards. 

As seen in Figure 5, FWDs’ generated pulse load lasts approximately 25 ms, reaching the highest load of ~53 kN after 15 ms from the initial start. In a larger time scale, to analyze strain versus time by the same FWD, 3 series with 3 drops in each series (with a 4 ms pause between each drop) are made, and more detailed response information can be seen in Figure 6, where one of those series is shown. There, an FBG strain sensor’s response data show the detected strain values when the horizontal distance between the FWD load plate center and the sensor is 50 cm (explained in more detail in the next subsection). With this distance between the optical strain sensor and the FWD’s load plate, for each drop, the average measured strain Δε is approximately 14 μm/m. Resonant oscillations are also observed after each drop (approximately for 0.5 s) until the momentum of the strain applied to the road infrastructure evens out. Parallel to the strain measurements, we also measure temperature by using the FBG optical sensor that is embedded into the cement-treated RAP mixture layer. The material temperature within the cement-treated RAP mixture layer, during experimental measurements, varied within a range of 1–1.5 °C and is measured by our embedded FBG temperature optical sensor.

Afterward, 15 separate FWD load drops (with 50 kN load each) are made on the top of the location where the FBG optical strain sensor is implemented. Based on the measured data, the distribution of received strain values made by FWD drops (see Figure 7) is analyzed. This is performed to be certain that the installment process ensures precise strain detection (and that of its values) under continuous load conditions during the time. As it also does on the road infrastructure, when real-time transport traffic passes the road pavement, leaving multiple strain-induced changes in a quick period.

On average, 34.63 μm/m for each of the series is estimated, and a strong correlation between the 15 sets of experimental FWD load data could be observed. In addition to that, the measured strain distribution of the FWD drops shows a Gaussian type of model. From what was previously mentioned, we can state that a stable operation of the SHM system model can be realized with the real-time transport traffic and the correct gathering of strain data, allowing for monitoring of both low- and high-intensity traffic over the road infrastructure.

### 4.1. Horizontal Distance from the FBG Optical Sensors Concerning FWD’s Load Plate Position

FWD experiments are further investigated by realizing different test scenarios. The acquired strain values are compared regarding the load positioning of the load plate concerning the embedded optical FBG sensor. Table 3 consists of all the data gathered when the horizontal distance between the central point of the fiber optical sensor and the FWD’s load plate is changed from the “on top of” position (0 cm away) to 50 cm with a step of 10 cm. This allows us to observe how large the difference is for the received average strain shift Δε_avg (μm/m) as well as the relative deviation (%) by the distance of the load plate on top of the road layer.

**Table 3 sensors-22-04581-t003:** Received strain measurement values induced by the FWD in its different positions regarding the implemented FBG optical strain sensor.

Horizontal distance between the FWD load plate center and FBG sensor	0 cm	10 cm	20 cm	30 cm	40 cm	50 cm
Δε_avg (μm/m)	34.63	30.21	28.16	23.13	19.71	14.03
Coefficient of variation (%)	4.26	6.64	3.28	5.24	5.59	6.26

As can be observed, the average strain shift Δε_avg is 34.63 μm/m in a scenario when the load plate is directly on the road’s top layer position (beneath of which is the embedded FBG optical strain sensor). However, as expected, this value decreases with the additional horizontal distance further away from the optical sensor, showing 14.03 μm/m when the load plate is 50 cm further away from the starting position. This means that a 50 cm large horizontal distance decreases the average detected strain shift value by approximately 20.60 μm/m compared to the highest (starting) value (34.63 μm/m). Moreover, as can be seen in Table 3, the coefficient of variation ranges from 3.28% up to 6.64%. From the observed data, we can state that the adjacent lane and traffic over it have a negligible effect on the measured strain data towards the lane where the FBGs are embedded. In this research, we studied the scenario where two lanes are built in both driving directions—thus, the road is wider and the impact from the adjacent traffic lane is even smaller.

The numerical data gathered and shown in Table 3 are also visually depicted in Figure 8. Here, a correlation between the received strain values (measured with the FBG optical strain sensor) and FWD’s load plate centrum distance position (according to the FBG optical sensors’ embedment position) is observed. In the graph, blue dots with lines represent the average strain values and the measurement variation range.

As can be seen in Figure 8, there is a strong linear correlation between the received strain shift values and the position distance of the FWD’s plate (concerning the FBG’s embedment). Thus, approving the proper installment of our FBG optical strain sensor and its correct operation, we can state that if an FBG optical strain sensor that has a sensitivity and protective coating as described in the previous section is implemented beneath the 220 mm of road infrastructure layers (SMA11—35 mm, AC22—60 mm, AC32—100 mm, cement treated RAP mixture layer—25 mm), then such a configuration will provide an accurate linear equation in regards to the sensor’s working range diapason.

### 4.2. Correlation between the Applied Stress and Measured Strain Values

Further into our research, to evaluate the precision of our previously mentioned real-time transport traffic strain measurement experiments (discussed in more detail in Section 5), further repeated tests are made (approximately half a year later from the time of the first experiments) by using the same FWD device. The temperature of the road’s cement-treated RAP mixture layer in these experiments is measured by our embedded FBG temperature’s optical sensor and is in the range of 24.8–25.1 °C. The effect of the temperature on the received strain amount is explained in more detail in Section 4.3. Here, we position the dropping plate of the FWD device right on the embedment position of the FBG strain optical sensor. Then, by changing the FWD’s applied stress values (from 470 kPa to 1170 kPa, with a step of ~200 kPa between the minimal and maximal values), we measure and record the received strain (μm/m) values. For each level of applied stress, 10 drops (5 times in a row, in 2 series) are made. In Figure 9, the received strain (μm/m) values at FWD’s generated stress levels (kPa) are shown, as the blue dots with lines represent the average strain and stress values, as well as the strain and stress measurement variation range.

As can be seen in Figure 9, a clear linear correlation can be observed between the received FBG optical strain sensor’s strain values (μm/m) and the FWD device’s induced stress (kPa) level. Numerically, 470 kPa of high stress leads to ~10.3 μm/m amount of strain, while 1170 kPa leads to ~23 μm/m. This means that previously gathered and measured data are accurate enough for SHM monitoring applications. No less important, this also proves that our embedment technique and methods (discussed in Section 3 and Section 4) can ensure the proper installment, longevity (at least half a year later, as shown in this research), and received data accuracy of such FBG strain optical sensor’s realization in road SHM applications.

### 4.3. The Effect of Temperature on the Received Strain Values

In this subsection, the received strain values at different temperatures (periods when measurements are taken) are analyzed and compared. The FWD device is positioned right on the embedment position of the FBG strain and temperature sensors. FWD load drops are fixed to 50 kN for each drop. The numerical data gathered aew shown in Table 4.

**Table 4 sensors-22-04581-t004:** Received strain measurement values induced by the FWD devices versus ambient temperature.

The temperature of the cement-treated RAP mixture layer during the experiments	1–1.5 °C	24.8–25.1 °C
Δε_avg (μm/m)	34.63	13.70
Coefficient of variation (%)	4.26	6.57

As can be seen in Table 4, with the temperature range of 1–1.5 °C (winter time), the received average relative strain value at the recycled layer is 2.5 times higher than at 24.8–25.1 °C (summertime). Such results can be explained by the temperature’s effect on the different layers of the road structure. As the environmental temperature decreases, so do the temperatures of the road’s layer materials. With lower temperatures, for example, during wintertime, the upper asphalt layers become stiffer, and thus the induced strain by the FWD device (or any other transport traffic vehicle) is more freely carried over to deeper layers of the road’s recycled layer. Thereby, a larger amount of the total strain is received at the FBG optical strain sensor that (in this scenario) is embedded in the cement-treated RAP mixture layer. When the environmental temperature has significantly increased (as shown in Table 4), so has the temperature in the upper layers of the road. Then, the upper layers of the road are not as stiff compared to the colder conditions, leading to more dissipation of the applied strain within those layers. Dissipation of the strain then can be observed as a smaller strain amount sensed directly with the FBG strain optical sensor.

## 5. Real-Time Transport Traffic Strain Monitoring Experiments

As experiments with the FBG strain optical sensor and FWD device approved the stable and correct operation, the next research phase consists of real-time transport traffic strain monitoring experiments. The temperature within the cement-treated RAP mixture layer during experimental measurements varies within a range of 1–1.5 °C. We have measured and recorded traffic that was passing over the road’s (specific location mentioned in the Section 3 of this article) pavement above the FBG’s embedment position. Moving vehicles there consisted of 2-axle passenger cars and up-to-6-axle heavy trucks. As there are different types of transportation passing the road, to better comprehend the strain-induced changes and their amount, we categorize all of the transport into four classes by the most common transport types passing the specific road. Those classes are: 2-axle passenger cars (Figure 10a), 2-axle minibuses (Figure 10b), 2–3-axle trucks (Figure 10c) and 4–6-axle trucks (Figure 10d–f). 

Figure 10. shows typical strain-induced changes (μm/m) in time (s) for all four classes of transport mentioned previously. As seen in Figure 10, an embedded FBG optical strain sensor allows for the precise detection of the strain-induced changes in time for transport with different amounts of axles. It is also possible to evaluate how much strain is induced from each axle of the passing vehicle. For instance, Figure 10a shows the strain measurements made by BMW X5 (distance between axles 2.975 m), where the measured time between the strain peaks (axle points) is 0.164 ms. Based on this data, we can calculate the movement speed of this car as follows: (3)$$v=\frac{s}{t}=\frac{2.975}{0.164}=18.14\left(\frac{m}{s}\right)=65.3\left(\frac{\mathrm{km}}{h}\right)$$

Thus, additionally, such realization of FBG optical sensors can provide information about the movement speed of the transport on the road where sensors are used. This valuable data can then be processed for various solutions—not only for the SHM but also to support fiber optic and Internet of Things applications, for instance, in traffic planning and road safety control. It is also important to state that due to the different amounts of induced strain by every axle of the specific vehicle, the relaxation time to reach the 0 strain level after the impact is also diverse. For some lighter (2-axle) vehicles, full relaxation to the zero micro-strain level might typically require up to an additional 3 s since the last contact, while for the heavier vehicles (5+-axle trucks) this is typically up to 5 s. Considering that the strain shift is determined as shown in Figure 8, then the strain-induced relaxation process has no adverse effect on the measurements regarding the induced strain by the next passing vehicle. Thus, it is important to highlight that with high-intensity traffic, the complete relaxation of the strain level might be observed rarely. However, there is a wide range of factors that affect the relaxation process and time. Some of the most important ones include the overall traffic intensity, type of the passing vehicle, its weight, load on each axle, passing trajectory (horizontal distance between the embedded sensor’s location and the vehicle’s tire) as well as the environmental (material’s) temperature.

Numerically, we also estimate the induced strain shift range (μm/m) of the previously mentioned transportations classes (Table 5).

All of the numerical values put in Table 5 show the strain-induced values for each axle of the specific mode of transportation as the road pavement and specific reference point on the road receive individual axle’s induced strain at one particular moment. However, as the mass of a specific vehicle depends on the brand and type of it, as well as the cargo carried, some of these values are clearly out of the typical measurement range (compared to other vehicles in their transport class) and are very unique. Thus, to better understand the more common strain-induced shift values, a typical strain (90% events) shift range (μm/m) is also calculated. For 2-axle passenger cars, it is 0.8–4.1 μm/m, for 2-axle minibuses 5.5–8.5 μm/m, for 2–3-axle trucks 11–26 μm/m, but for 4–6-axle trucks it is 14–36 μm/m per each axle.

For comparison, as seen in Table 5, the axle of the passenger car typically induces approximately 2–10 times less strain than the axle of the 2-axle minibus. Meanwhile, typically, axles of the passenger car induces approximately 3–45 times less strain than the axle of a heavy 4–6-axle truck.

## 6. Conclusions

Pavement SHM applications completed by realizing fiber optical sensors not only assists in making appropriate decisions for planning maintenance and reconstruction works but also validates the proposed design solution. Such an aspect is gaining increased importance since new materials and structural layers are introduced under changing traffic volumes and road loading conditions.

This research approved the successful integration of the FBG optical strain and temperature sensors that were embedded during the construction of a road of national importance. FBG sensors were integrated within the permanent infrastructure—into the cement-treated RAP mixture layer (220 mm deep). Short-term and long-term experiments with the FWD device and real-time transport traffic were carried out over half a year. We calculated and concluded that the received strain levels in this particular configuration are dependent upon various factors. The major ones are the horizontal distance from the embedment position of the optical strain sensor and strain inducer (FWD or real-time transport traffic in this research), the amount and intensity of the load that induces strain, and last but not least—the temperature of the environment (road layers). 

As described in the article, by using the FWD, we concluded that the received strain amount decreases with the additional distance further away from the optical sensor, showing a 14.03 μm/m large strain when the load plate is 50 cm further away from the embedment position. This means that a 50 cm large horizontal distance decreased the average detected strain shift value by approximately 20.60 μm/m compared to the highest (starting) value (34.63 μm/m) recorded on top of the embedment position. In addition, by changing the applied stress values of the FWD (from 470 kPa to 1170 kPa), we concluded that 470 kPa high stress leads to ~10.3 μm/m for the amount of received strain, while 1170 kPa leads to ~23 μm/m when the stress is induced on top of the embedment position. As for the real-time transport traffic—2-axle passenger cars generated 0.8–4.1 μm/m large strain per axle, 2-axle minibuses 5.5–8.5 μm/m, 2–3-axle trucks 11–26 μm/m, but 4–6-axle trucks14–36 μm/m per each axle. The axle of a passenger car induced approximately 2–10 times less strain than the axle of a 2-axle minibus. Meanwhile, the axle of the passenger car induces approximately 3–45 times less strain than the axle of a heavy 4–6-axle truck. As for the recycled layer’s temperature factor, we estimated that the temperature range of 1–1.5 °C (winter time) leads to the received average relative strain values that are 2.5 times higher than the ones recorded when the temperature is ranging between 24.8–25.1 °C (summertime). As explained in the article, different temperatures determine the stiffness of the road’s upper layers, and thus also the number of received strain values.

The sensing setup developed, the methods described and used, and the components utilized can be realized for further large-scale FBG optical strain and temperature realizations within the construction of road infrastructure, thus providing for the capability to use, study and develop the road SHM industry as well as other spheres where such solutions are appealing.

## Figures and Tables

**Figure 1 sensors-22-04581-f001:**
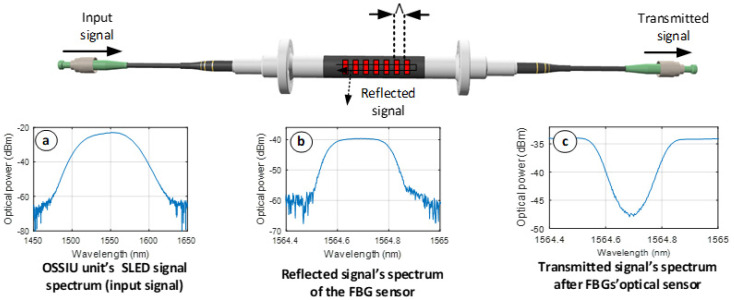
The characteristic and measured spectrum of commercial [23] FBG strain sensor.

**Figure 2 sensors-22-04581-f002:**
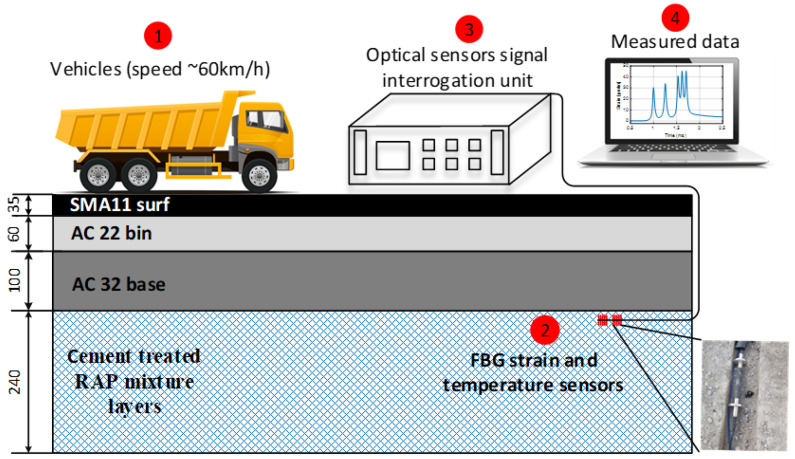
Schematic setup of the FBG optical sensors’ integration (mm deep) and placement for strain measurement experiments.

**Figure 3 sensors-22-04581-f003:**
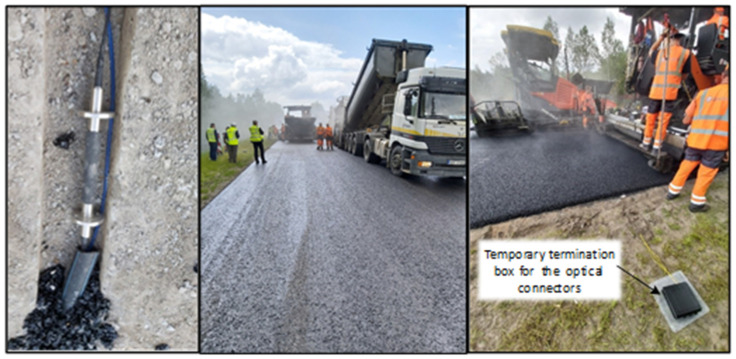
Photos from the construction process of road infrastructure where FBG sensors are integrated.

**Figure 4 sensors-22-04581-f004:**
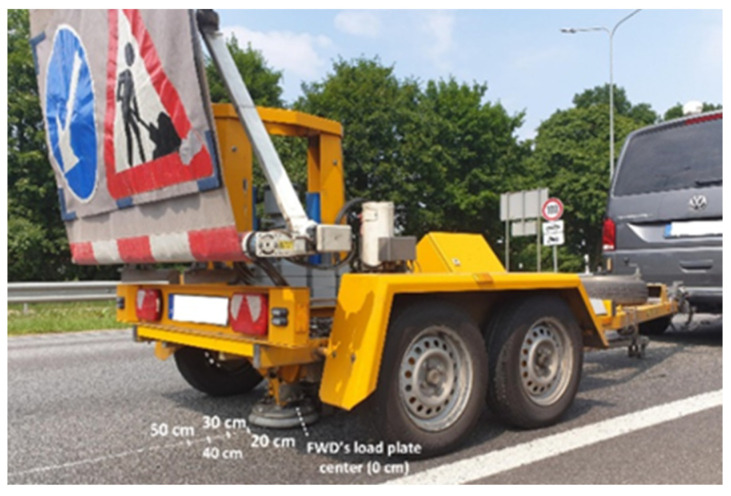
Realization of FWD experiments for calibration purposes of the embedded optical sensors.

**Figure 5 sensors-22-04581-f005:**
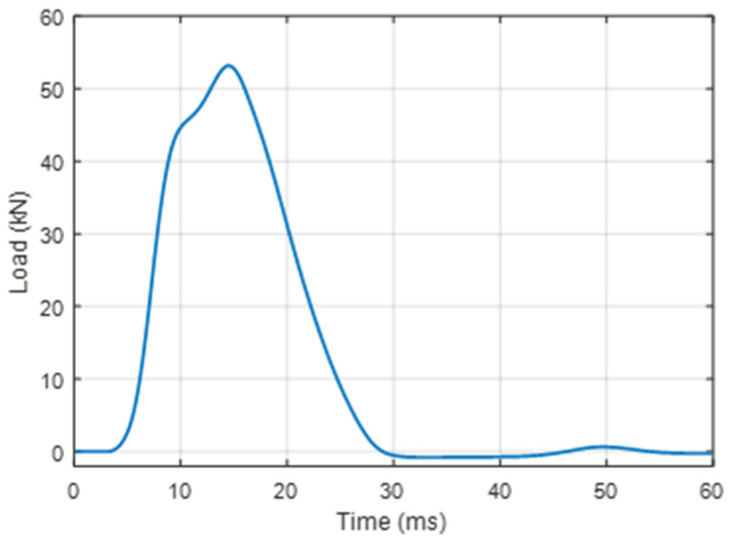
Measured pulse load that is induced by the FWD.

**Figure 6 sensors-22-04581-f006:**
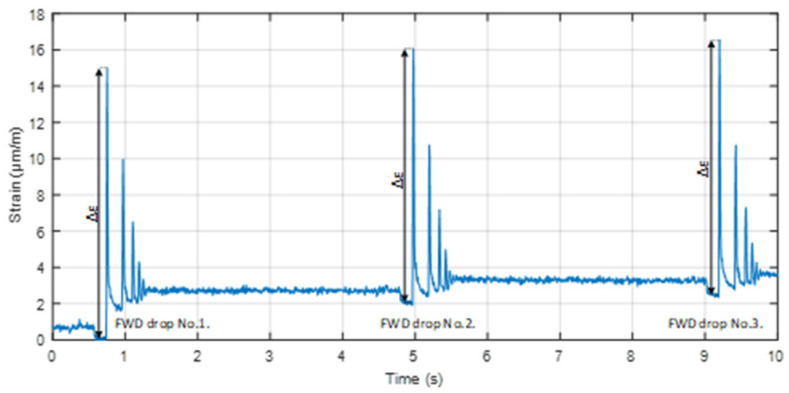
Monitoring of the strain versus time induced by FWD (3 drops) when the horizontal distance between the center of the FWD load plate and the embedment position of the optical sensor is 50 cm.

**Figure 7 sensors-22-04581-f007:**
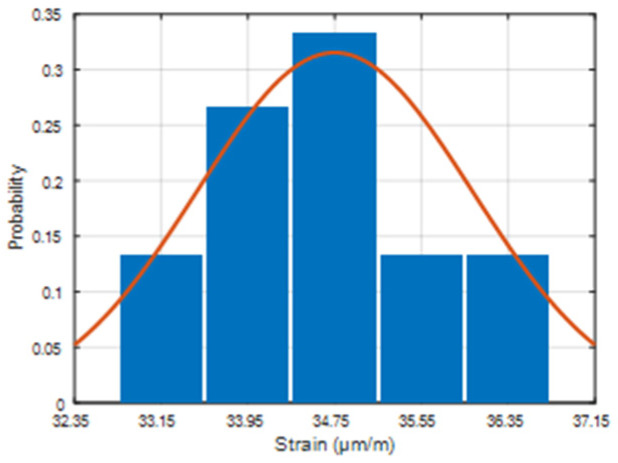
FBG sensor’s measured strain distribution of FWD drops when the FWD load plate is located on top of (directly on) the position under where the FBG strain sensor is embedded.

**Figure 8 sensors-22-04581-f008:**
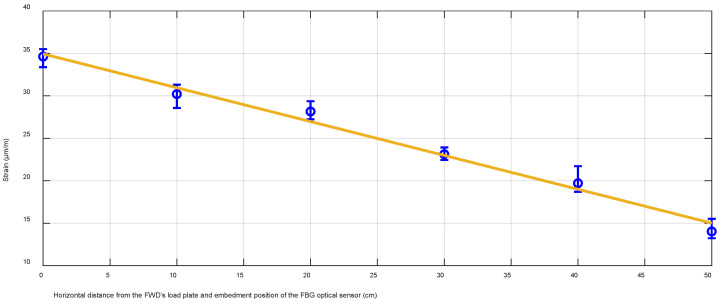
Correlation between the received strain values (measured with FBG strain optical sensor) and FWD’s load plate centrum distance position (according to the FBG optical sensors’ embedment position).

**Figure 9 sensors-22-04581-f009:**
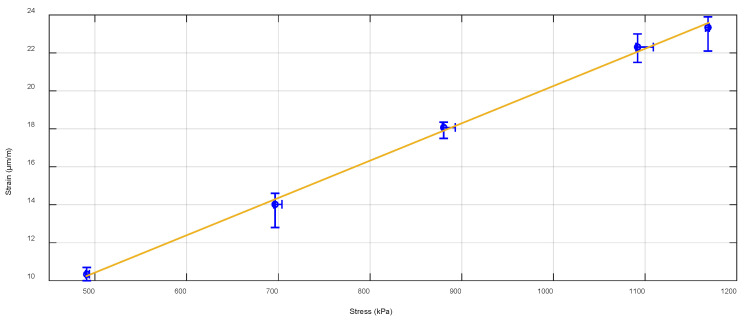
Received average FBG optical strain sensor’s strain values (μm/m) for every induced stress (kPa) level made by the FWD device.

**Figure 10 sensors-22-04581-f010:**
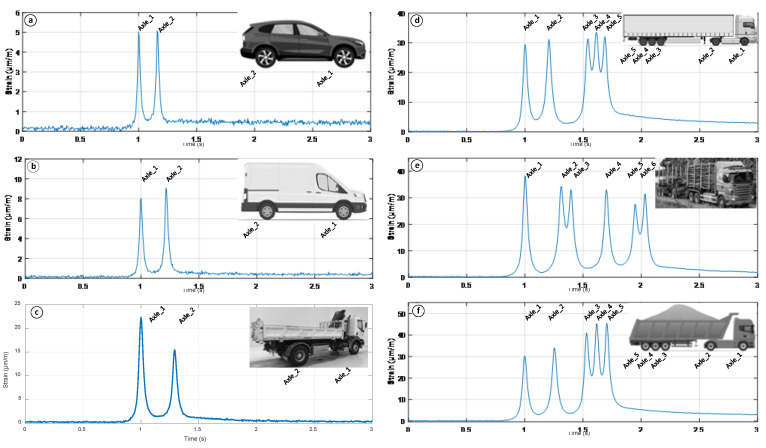
Real-time transport traffic strain measurements and visual data acquired from vehicles passing over the surface of the road where the FBG optical sensor was integrated.

**Table 1 sensors-22-04581-t001:** The main parameters of the developed and used optical sensors signal interrogation unit.

Parameters	Value
Wavelength operation band	1520–1575 nm
Max scan frequency for single optical channel	10 kHz
Wavelength resolution	1 pm
Remote maximum operation distance	40 km

**Table 2 sensors-22-04581-t002:** The main parameters of the FBG temperature and strain sensor.

FBG No.	Measured Parameter	Central Wavelength	Accuracy	Measurement Range
1	Strain	1544.800 nm	<1 µm	±5000 µɛ
2	Temperature	1554.558 nm	±0.3 °C	−40–120 °C

**Table 5 sensors-22-04581-t005:** Measured induced strain shift range and calculated typical strain shift of real-time transport traffic.

	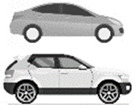	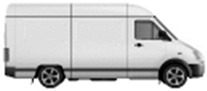	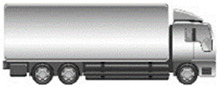	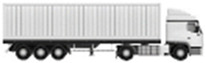
**Induced strain shift range (μm/m)**	0.7–6	4.6–13	10–38	12–42
**Typical (90% event) strain shift range (μm/m)**	0.8–4.1	5.5–8.5	11–26	14–36

## Data Availability

The data used to support the findings of this study are available from the first author upon request.

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
