# Peer review of "Road Pavement Structural Health Monitoring by Embedded Fiber-Bragg-Grating-Based Optical Sensors"

_sensors, 2022, doi:10.3390/s22124581_

Round 1
Reviewer 1 Report
Line 35: what does it means E10? Please specify.
Line 46: the authors may also consider the paper published by Montanini on Review of Scientific Instruments vol. 78(2007) which deals with an FBG-based optical psychrometer.
Line 82: please provide a reference or specify the manufacturer
Line 190-191: “…and a strong correlation between the data could be observed” what do you mean for strong correlation? Between which experimental data? Perhaps the authors would mean that experimental data follow a Gaussian model? Please clarify.
Line 203: please delete “and its deviation”
Line 219: “but the error bars - the strain variation range.” Sounds strange….
Line 246: “but the error bars - strain and stress variation range.” Sounds strange….
Reviewer 2 Report
In their work, the authors perform monitoring of the pavement using a fiber-optic sensor. The advantage of the work is that the conditions of the sensor are close to the real one. In particular, the authors were able to measure load patterns when different types of vehicles move on the road. I can recommend the paper for publication after the authors have taken into account the following brief remarks:
1. In Figure 12, you can see that the sensor readout does not return to its original value. If the load was about 0 microstrength at the beginning, after 2-3 seconds it may be about 2-3 microstrengths. It would be useful to discuss the relaxation of the sensor and indicate typical times.
2. The authors investigated fairly simple cases of exposure to the transducer. It would be useful to briefly discuss the more complex cases - the effect on the sensor of traffic from the adjacent lane, the operation of the sensor in fast traffic flow, and other potential limitations in use.
Reviewer 3 Report
The manuscript demonstrated an application of FBG sensors, which is interesting to the community of Structural Health Monitoring. It could be accepted for publication. It will be more useful to demonstrate the priority of FBG sensors or the problems during the application. If possible, the comparison between the results of the traditional electrical sensors and FBG sensors is preferred.
